# Heart-Rate-to-Blood-Pressure Ratios Correlate with Malignant Brain Edema and One-Month Death in Large Hemispheric Infarction: A Cohort Study

**DOI:** 10.3390/diagnostics13152506

**Published:** 2023-07-27

**Authors:** Xindi Song, Yanan Wang, Wen Guo, Meng Liu, Yilun Deng, Kaili Ye, Ming Liu

**Affiliations:** 1Department of Neurology, West China Hospital, Sichuan University, Chengdu 610041, China; xindi_song@foxmail.com (X.S.); yanan_wang1105@163.com (Y.W.); guowen@wchscu.cn (W.G.); liumeng2142001303@163.com (M.L.); deng.yilun@hotmail.com (Y.D.); kl_ye2023reg@163.com (K.Y.); 2Center of Gerontology and Geriatrics, West China Hospital, Sichuan University, Chengdu 610041, China; 3Department of Neurology, No. 3 People’s Hospital of Chengdu, Chengdu 610031, China

**Keywords:** malignant brain edema, shock index, blood pressure, heart rate, biomarkers

## Abstract

Introduction: Large hemispheric infarction (LHI) can lead to fatal complications such as malignant brain edema (MBE). We aimed to investigate the correlation between heart-rate-to-blood-pressure ratios and MBE or one-month death after LHI. Methods: We prospectively included LHI patients from a registered cohort. Hourly heart-rate-to-blood-pressure ratios were recorded as a variation of the traditional shock index (SI), SI_s_ and SI_d_ (systolic and diastolic pressures, respectively), and calculated for mean and variability (standard deviation) in 24 h and two 12 h epochs (1–12 h and 13–24 h) after onset of symptoms. MBE was defined as neurological deterioration symptoms with imaging evidence of brain swelling. We employed a generalized estimating equation to compare the trend in longitudinal collected SI_s_ and SI_d_ between patients with and without MBE. We used multivariate logistic regression to investigate the correlation between SI_s_, SI_d_ and outcomes. Results: Of the included 162 LHI patients, 28.4% (46/162) developed MBE and 25.3% (40/158) died within one month. SI_s_ and SI_d_ increased over baseline in all patients, with a similar ascending profile during the first 12 h epoch and a more intensive increase in the MBE group during the second 12 h epoch (*p* < 0.05). During the overall 24 h, patients with greater SI_d_ variability had a significantly increased MBE risk after adjustment (OR 3.72, 95%CI 1.38–10.04). Additionally, during the second 12 h epoch (13–24 h after symptom onset), patients developing MBE had a significantly higher SI_d_ level (OR 1.18, 95%CI 1.00–1.39) and greater SI_d_ variability (OR 3.16, 95%CI 1.35–7.40). Higher SI_d_ and greater SI_d_ variability within 24 h independently correlated with one-month death (all *p* < 0.05). Within the second 12 h epoch, higher SI_s_, higher SI_d_ and greater SI_d_ variability independently correlated with one-month death (all *p* < 0.05). No significant correlation was observed in the first 12 h epoch. Conclusions: Higher and more fluctuated heart-rate-to-blood-pressure ratios independently correlated with MBE development and one-month death in LHI patients, especially during the second 12 h (13–24 h) epoch after onset.

## 1. Introduction

Stroke is the leading cause of death and disability worldwide [1,2], and the most common type of stroke is acute ischemic stroke [2]. Large hemispheric infarction (LHI) is the very ischemic stroke subtype with the poorest prognosis [3]. LHI patients may develop rapidly progressing malignant brain edema (MBE), likely resulting in an impaired state of consciousness, herniation, severe independence or death [4]. The fatality of MBE under conservative intensive care is around 80% [5,6], with medication or surgery having limited effectiveness [3,5]. Therefore, it is necessary to prevent the occurrence of MBE to reduce fatality and improve the prognosis of LHI.

The autoregulation of cerebral blood flow is injured after stroke, increasing the dependence of cerebral blood supply on systemic blood pressure [7,8]. The formation of MBE mainly results from disruption of the blood–brain barrier, which physiologically correlates with blood pressure [4,5,9]. Some previous studies explored the correlation between blood pressure and prognosis of general stroke [10,11], but investigation into LHI patients is limited. Moreover, few studies have examined the impact of hemodynamic indexes that physiologically link to blood pressure, for example, the heart rate.

The shock index (SI), calculated as the heart rate divided by systolic blood pressure, is a commonly used index in intensive care [12]. Recent studies showed that admission SI at extremities appeared to predict poor outcomes in general stroke [13,14]. Here, we hypothesize that SI may correlate with MBE or related fatality in LHI patients. No study has explored this correlation. Moreover, alteration of HR affects diastolic blood pressure more significantly than systolic pressure, and diastolic pressure takes up a greater proportion of the mean artery pressure, which determines cerebral blood flow. It is reasonable to suspect that the heart-rate-to-diastolic-pressure rate may correlate with, if not more significantly than the traditional shock index, MBE or related fatality after LHI.

The natural history of brain edema can be categorized into cytotoxic and vasogenic edema. Cytotoxic edema is defined as cell swelling caused by intracellular fluid accumulation and is initially observed within hours after stroke onset and then declined within 1 day. Vasogenic edema is caused by extracellular fluid accumulation resulting through disruption of the blood–brain barrier and is usually developed within two to three days after onset and maintained for several days [9]. Therefore, the influence of heart-rate-to-blood-pressure ratios may vary at different recording epochs. 

In this research, we aimed to explore the correlation between heart-rate-to-blood-pressure ratios and MBE, as well as one-month death after LHI. 

## 2. Materials and Methods

### 2.1. Study Population

We enrolled patients with acute ischemic stroke consecutively admitted between January 2020 and December 2021 from our registry cohort database [15,16]. Acute ischemic stroke was diagnosed according to established guidelines [17]. LHI was defined as infarction with visible hypodensity over a third of the middle cerebral artery (MCA) territory in computerized tomography and/or magnetic resonance imaging within 6 h of onset, or over half the territory of MCA within 48 h of onset [18]. All patients completed computerized tomography on admission. Follow-up brain imaging was performed within 7 days of admission or when clinical worsening occurred. LHI adult patients admitted within 24 h of onset were included. Patients were excluded if they: (1) had bilateral ischemia, (2) had recurrent stroke, (3) had parenchymal hemorrhage type 2 [19] occurring before or at follow-up imaging, and/or (4) had no blood pressure or heart rate measurements. 

### 2.2. Data Collection

We collected demographic and clinical data including age, gender, body temperature, National Institute of Health stroke scale (NIHSS) and Glasgow Coma Scale (GCS) on admission, hypertension, diabetes mellitus, hyperlipidemia, atrial fibrillation and stenosis/occlusion of intra/extracerebral arteries; ischemic area (whether invisible hypodensity ≥ 1/2 territory of MCA), Trial of ORG 10,172 in Acute Stroke Treatment (TOAST) classification [20]. Treatments including thrombolysis, thrombectomy, and dehydration treatment were documented. Any in-hospital infection, including pneumonia, urinary infection or others, was also recorded.

### 2.3. Heart-Rate-to-Blood-Pressure Ratios Measurement

We collected repeated time-stamped blood pressure and heart rate of all included patients within the first 24 h post-onset. Blood pressure and heart rate were measured simultaneously by noninvasive monitoring devices and were recorded by trained nurses hourly (±30 min) after admission. The relationship between heart rate and diastolic blood pressure can be represented as a variation of the traditional shock index, wherein heart-rate-to-blood-pressure ratios are assigned as SI_s_ and SI_d_ for systolic and diastolic pressures, respectively. The equations for SI_s_ and SI_d_ are derived by dividing heart rate by systolic or diastolic blood pressure. Hourly SI_s_ and SI_d_ were recorded and calculated for mean, maximum, minimum, range (difference between maximum and minimum), standard deviation (SD) and coefficient of variation (CV) for the entire recording epoch (24 h) and during two 12 h epochs (1–12 h and 13–24 h after onset). For patients who developed MBE within 24 h of onset, the heart-rate-to-blood-pressure ratios collected after MBE development were excluded for analysis in order to ensure the appropriate sequential order of studied parameters and outcome.

### 2.4. Outcomes

The primary outcome was MBE, defined as midline shift ≥ 5 mm with a deteriorated status of consciousness or anisocoria, or indications for decompressive craniectomy. All images were independently viewed by two researchers (X.S and W.G) who were blinded to clinical data. A final decision was made by a senior neurologist (M.L) if two researchers could not reach a consensus. Presence of mild degree of hemorrhagic transformation was permitted, since it was not considered to result in mass effect [21]. Secondary outcomes were one-month death and three-month poor functional outcome (modified Rankin scale (mRS) ≥ 3). Patients or, if not possible, their relatives were followed up by telephone interview at one and three months after stroke. Failure to contact after three different attempts was recorded as lost to follow-up.

### 2.5. Statistical Analysis

All data were reported as mean ± standard deviation (SD), median (interquartile range, IQR) or count (percentage). Inter-group differences were compared by χ2 test or Fisher’s exact test for categorical variables, and Student’s *t*-test or Mann–Whitney U test for continuous variables as appropriate. We employed the generalized estimating equation (GEE) to compare the trend in longitudinal collected SI_s_ and SI_d_ measurement values between patients with and without MBE, as well as between patients who survived and those who died within one month. This method allowed us to account for the correlation between the repeated measurements over time, improving the accuracy of our parameter estimates and providing scientific rigor to our analysis. We used multivariable logistic regression to analyze independent correlation between SI_s_ or SI_d_ parameters and outcomes. SD and CV values of SI_s_ and SI_d_ were entered into the regression model as log-transformed values due to the skewness distribution of data, while other SI_s_ and SI_d_ parameters were entered as ten-fold values (per 0.1-unit increase). We preselected the adjusted covariates with clinical significance for MBE (age, body temperature, NIHSS, ischemic area, hypertension, atrial fibrillation, dehydration treatment, and in-hospital infection), and one-month death/three-month poor outcome (age, body temperature, NIHSS, ischemic area, hypertension, atrial fibrillation, and in-hospital infection). Cumulative risk of one-month death was estimated with Kaplan–Meier curves and compared by log-rank test. We further conducted a restricted cubic spline regression model to assess the non-linear dose effect of SI_s_ or SI_d_ parameters on outcomes with four knots (at the 5th, 35th, 65th, and 95th percentiles) [22]. We determined the cutoff of SI_s_ or SI_d_ that best predicted MBE and one-month death based on receiver operating characteristic curves and the Youden index. All statistical analyses were performed by SPSS (version 26.0; IBM, Chicago, IL, USA), STATA 16.0 (Corporation, College Station, TX, USA), and GraphPad Prism (version 8.01; GraphPad Software, San Diego, CA, USA). A two-sided *p* < 0.05 was considered statistically significant.

## 3. Results

### 3.1. Baseline Characteristics of Patients

During the study period, 175 LHI patients were preliminarily screened. A total of 13 patients were excluded: 3 patients developed bilateral ischemia; 1 patient was diagnosed with recurrent stroke at this visit; 6 patients developed parenchymal hemorrhage type 2 before imaging evaluation for MBE; 3 patients were with absence of blood pressure and heart rate measurements (Appendix A). We finally enrolled 162 LHI patients (43.8% males, mean age 70.3 years old) with a median NIHSS of 16 (IQR 12–21). The median delay from onset of symptoms to admission was 5 h (IQR 3–12). Of 162 included LHI patients, 28.4% (46/162) developed MBE. The median interval between onset and MBE was 36.3 h (IQR 24.3–59.4), among which 85% (39/46) with interval over 24 h. A total of 2.5% patients (4/162) at one month and 4.3% patients (7/162) at three months were lost to follow-up. A total of 25.3% (40/158) died within one month, and 77.4% (120/155) had a three-month poor functional outcome.

### 3.2. Temporal Evolution of Heart-Rate-to-Blood-Pressure Ratios

We included 2138 sets of time-stamped blood pressure and heart rate measurements in total. For up to 24 h after onset, 14 ± 6 sets of blood pressure and heart rate were recorded per patient, 6 ± 3 and 8 ± 3 for the first and second 12 h epoch, respectively. Missing data were mainly due to pre-hospital referral and transferring for thrombectomy or follow-up imaging.

Temporal evolution of SI_s_ and SI_d_ in LHI patients with and without MBE are shown in Figure 1. SI_s_ and SI_d_ increased in the early phase (first 12 h epoch, 1–12 h after symptom onset) for both the MBE group and non-MBE group, then reached a plateau for the remainder of the monitoring epoch in the non-MBE group. In contrast, SI_s_ and SI_d_ continued to rise at a lower rate in the MBE group, beginning at approximately 13 h after onset. During the second 12 h epoch (13–24 h after symptom onset), in patients with MBE, SI_s_ and SI_d_ were higher than that in patients without MBE (*p* < 0.001 for SI_s_, *p* = 0.036 for SI_d_ by GEE method for MBE). We did not find differences in SI_s_ and SI_d_ during the entire 24 h recording period or during the first 12 h epoch between patients with and without MBE (all *p* > 0.05 for SI_s_ and SI_d_ by GEE method for MBE). The temporal evolutions of SI_s_ an SI_d_ were similar in patients who died or had survived at one month (Appendix A).

### 3.3. Correlation of Heart-Rate-to-Blood-Pressure Ratios with Malignant Brain Edema

The clinical features and heart-rate-to-blood-pressure ratio parameters in patients with and without MBE were compared in Table 1 and Table 2. Patients with MBE were more likely to have an ischemic area over half the territory of MCA (78.26% vs. 47.41%, *p* < 0.001) than those without MBE. Patients with MBE had a greater SI_d_ variability during the entire 24 h recording epoch (SI_d_ SD 0.23 bpm/mmHg vs. 0.19 bpm/mmHg, *p* = 0.017), and higher SI_d_ level (1.26 bpm/mmHg vs. 1.13 bpm/mmHg, *p* = 0.028) with greater variability (SI_d_ SD 0.20 bpm/mmHg vs. 0.16 bpm/mmHg, *p* = 0.018) during second 12 h recording epoch (13–24 h after onset). 

In Table 3, during the overall 24 h post-stroke epoch, a trend in higher SI_d_ level was observed in the MBE group (OR 1.19, 95%CI 0.98–1.44, *p* = 0.074), though no significant correlation was observed after adjustment. In contrast, a higher maximum SI_d_ value (OR 1.15, 95%CI 1.04–1.28, *p* = 0.006), wider SI_d_ range (OR 1.09, 95%CI 1.00–1.18, *p* = 0.043) and greater SI_d_ variability (OR for log-SD 3.72, 95%CI 1.38–10.04, *p* = 0.010) during the overall 24 h recording epoch significantly correlated with MBE development after adjustment. 

The same effect direction with a stronger correlation was observed during the second 12 h epoch (13–24 h after onset). During this epoch, a higher SI_d_ level (OR 1.18, 95%CI 1.00–1.39, *p* = 0.046) independently correlated with an increased risk of MBE after adjustment. Additionally, 13–24 h after onset, a higher maximum SI_d_ value (OR 1.14, 95%CI 1.03–1.26, *p* = 0.012), wider SI_d_ range (OR 1.18, 95%CI 1.05–1.32, *p* = 0.006) and greater SI_d_ variability (OR for log-SD 3.16, 95%CI 1.35–7.40, *p* = 0.008) independently correlated with an increased risk of MBE. No SI_d_ parameters in the first 12 h epoch presented significant correlation with MBE after adjustment. No significant correlation between SI_s_ parameters with MBE were observed.

According to restricted cubic spline regression, during the second 12 h epoch after onset, the correlation between the mean SI_d_ level and MBE were in a non-linear dose–effect manner (Figure 2). The Youden index identified a mean SI_d_ level of 1.11 bpm/mmHg during the second 12 h epoch after onset (13–24 h after onset) as able to discriminate between patients with and without MBE. The area under the receiver operating characteristic curve was 0.61 bpm/mmHg, sensitivity was 78%, and specificity was 47%.

### 3.4. Correlation of Heart-Rate-to-Blood-Pressure Ratios with Secondary Outcomes

Appendix A shows the clinical features for patients who died or survived within one month. Patients who died within one month had greater variability of heart-rate-to-blood-pressure ratios than those who survived within one month during the overall 24 h recording epoch (all *p* < 0.05, Appendix A). During the 13–24 h epoch, higher SI_s_ (0.71 bpm/mmHg vs. 0.64 bpm/mmHg, *p* = 0.049) and SI_d_ (1.26 bpm/mmHg vs. 1.13 bpm/mmHg, *p* = 0.013) levels were observed in patients who died within one-month (Appendix A).

In Table 3, during the overall 24 h recording epoch, a higher SI_d_ level (OR 1.23, 95%CI 1.01–1.50, *p* = 0.037) independently correlated with one-month death after adjustment. Moreover, a higher maximum SI_d_ value (OR 1.17, 95%CI 1.05–1.29, *p* = 0.003), wider SI_d_ range (OR 1.18, 95%CI 1.06–1.31, *p* = 0.003) and greater SI_d_ variability (OR for log-SD 3.45, 95%CI 1.28–9.28, *p* = 0.014) during overall 24 h independently correlated with increased risk of MBE. 

No SI_s_ or SI_d_ parameters in the first 12 h epoch presented a significant correlation with one-month death. During the second 12 h epoch, a higher SI_s_ level (OR 1.28, 95%CI 1.00–1.65, *p* = 0.049) independently correlated with one-month death. Meanwhile, a higher SI_d_ level (OR 1.25, 95%CI 1.06–1.48, *p* = 0.009), a higher maximum SI_d_ value (OR 1.16, 95%CI 1.05–1.28, *p* = 0.004), a wider SI_d_ range (OR 1.18, 95%CI 1.05–1.32, *p* = 0.004) and greater SI_d_ variability (OR for log-SD 2.38, 95%CI 1.04–5.43, *p* = 0.039) during this epoch independently correlated with one-month death. The cumulative incidences of one-month death were significantly higher among patients with higher SI_s_ or SI_d_ levels during the 13–24 h epoch (Appendix A). The Youden index identified a mean SI_s_ of 0.65 bpm/mmHg and SI_d_ level of 1.15 bpm/mmHg 13–24 h after onset as able to discriminate between patients who died or survived within one month. The area under the receiver operating characteristic curve for SI_s_ and SI_d_ was 0.63 bpm/mmHg (sensitivity 72%, specificity 53%) and 0.64 bpm/mmHg (sensitivity 72%, specificity 56%), respectively. 

No difference in SI_s_ or SI_d_ parameters was observed between patients with and without a three-month poor functional outcome (all *p* > 0.05).

## 4. Discussion

To our knowledge, this study is the first to examine the temporal evolution of heart-rate-to-blood-pressure ratios (SI_s_ and SI_d_), as well as their impact on MBE and one-month death. We are also the first to investigate this correlation among specific, fatal stroke subtypes as LHI patients, using a registry cohort database. In this study, SI_s_ and SI_d_ increased over baseline in all patients, with similar profiles during the first 12 h epoch and a more intensive increase in patients with MBE or one-month death during the second 12 h epoch. Higher and more fluctuated heart-rate-to-blood-pressure ratios independently correlated with increased risk of MBE and one-month death in LHI patients, especially during the second 12 h (13–24 h) epoch after onset.

The median interval from onset to MBE development in our study was 36.3 h, which corresponded to previous studies [9], and 85% (39/46) of MBE was developed with an interval over 24 h. For the seven patients who developed MBE within 24 h of onset, we did not apply the heart-rate-to-blood-pressure ratios collected after MBE development for analysis, in order to ensure the appropriate sequential order of studied parameters and outcomes. For the remaining patients, we included all collected heart-rate-to-blood-pressure ratios for analysis. Therefore, it is much more likely that the MBE development was influenced by ratio changes and not the other way around, since for each individual patient, the studied ratios were collected strictly before MBE occurred. We failed to conduct sensitivity analysis by excluding those who developed MBE within 24 h of onset due to limited study samples, which needs to be validated in a further enlarged cohort. 

Our study revealed a significant correlation between the variability of continuous measurement of SI_d_ levels within 24 h after onset and the occurrence of MBE. Additionally, we found a significant correlation between an increase in the mean and variability of SI_d_ levels during the 13–24 h epoch after onset and the occurrence of MBE. The relationship between heart-rate-to-blood-pressure ratios and one-month mortality showed similar significance, that higher SI_d_ with greater variability is associated with one-month mortality, particularly during the 13–24 h epoch after onset. Higher SI_s_ during the 13–24 h epoch after onset was also independently correlated with one-month mortality. Our result is in agreement with previous studies. Two studies have investigated the correlation between SI_s_ and stroke outcome. McCall suggested that admission SI_s_ at both high and low extremities may predict short-term fatality among general stroke [13]. According to Myint, higher SI_s_ at presentation to emergency predicts patient-related clinical outcomes in general stroke [14]. Of note, previous studies only recorded the first admission SI_s_ of unselected patients, including both ischemic and hemorrhagic stroke populations. Our result specifically focused on LHI patients, who are more likely to develop MBE, thus needing more intensive monitoring. Meanwhile, we investigated the impact of both the absolute level and variability of the heart-rate-to-blood-pressure ratio by serially collecting them for 24 h after onset. Moreover, compared to previous studies, we extended the traditional shock index by adding a variation of it, SI_d_. Diastolic pressure has a greater impact than systolic pressure on the mean arterial pressure, which is the most commonly used target regarding cerebral blood flow. In our research, the SI_d_ parameter showed a significant correlation with MBE, while the SI_s_ parameter did not, suggesting that in LHI patients, diastolic blood pressure may play a more critical role than systolic blood pressure in brain edema development. This finding suggests that more attention should be paid to diastolic blood pressure control when preventing MBE development in clinical practice.

The mechanism underpinning differences in SI_s_ and SI_d_ remains unknown. No difference in systolic and diastolic blood pressure was observed between groups. Previous research reported an increased incidence of arrhythmias, or changes in the electrocardiogram in stroke [23,24,25,26,27]. Ischemic or hemorrhagic cerebral damage may impair the function of the autonomic nervous system and involves a higher sympathetic activity and a reduction in vagal tone, leading to an autonomic imbalance in heart rate regulation, most of which is tachyarrhycardia [27,28,29,30]. The risk of post-stroke tachyarrhycardia increases proportionally with stroke severity [31,32]. Therefore, LHI patients may feature a higher risk of post-stroke tachyarrhycardia, which correlate with poor clinically related outcomes [29,32,33]. HR was higher in patients who died within one month, although the difference was not statistically significant after adjustment for covariables (data now shown). This finding partially explained the impact of SI_s_ and SI_d_ on poorer outcomes, while also suggesting that heart rate alone may not independently influence the outcome. 

Another underlying mechanism for the association between elevated heart-rate-to-blood-pressure ratios and MBE as well as one-month mortality after acute stroke might be due to abnormalities in autonomic control, such as decreased baroreflex sensitivity (BRS). BRS is an index that quantifies the extent of baroreflex modulation of the heart rate in response to changes in blood pressure [34]. Previous studies reported a reduction in BRS in stroke patients [35], which is linked to an imbalance in autonomic regulation with a decline in the function of the parasympathetic nervous system and enhanced sympathetic activity [36] Sympathetic activation following a stroke can cause an increase in blood pressure while reduced BRS leads to weakened regulation of the heart rate in response to changes in blood pressure, leading to an insufficient decline in the heart rate and higher heart-rate-to-blood-pressure ratios. Moreover, BRS is inversely correlated with blood pressure variability [37], and impaired BRS can lead to an increase in blood pressure variability, which may also increase the variability of heart-rate-to-blood-pressure ratios. Impaired BRS appears to be related to brain hypoperfusion [38], thus leading to MBE development. Additionally, impaired BRS was reported to be associated with increased long-term mortality after acute ischemic stroke [36]. 

Elevated SI_s_ is reported to be of prognostic importance for infection, which may influence stroke outcomes [12]. Patients diagnosed with LHI have a relatively high incidence of in-hospital infection, with the pneumonia rate estimated to be between 39% and 54% [39,40] and urinary incontinence around 18.4% [39]. LHI patients with poor functional outcomes have a considerably higher incidence of pneumonia (67.1%) and urinary incontinence (27.9%) [39]. This phenomenon may be attributed to various factors, such as impaired consciousness post-stroke, aspiration, and prolonged hospitalization [41]. In this study, the rate of any in-hospital infection rate was 77.6%. The higher infection rate may be due to older age (70 vs. 61 years) and higher NIHSS (16 vs. 14) when compared to the previous study population [39]. It is also important to note that the COVID-19 pandemic may have contributed to an increase in in-hospital infection rates since patients in our study were included during this specific period. As a result, we included in-hospital infection as a covariate in our analysis to account for this confounding factor, and still identified a prognosis significance of elevated SI_s_ or SI_d_ for MBE and one-month death. However, due to a limited sample size, we regret that we were unable to perform a sensitivity analysis on patients without in-hospital infections. Therefore, further research involving larger sample sizes is needed to explore this issue further.

We investigated the association between the heart-rate-to-blood-pressure ratio parameters and the natural history of MBE progression by dividing the overall 24 h recording epoch into two 12 h epochs. The progression of MBE can be separated into two stages: cytotoxic edema and vasogenic edema. The initial stage of cytotoxic edema occurs within the first few hours and declines within one day, which aligns temporally with the first 12 h epoch. The second stage of vasogenic edema develops within one to three days and is maintained for several days [9], which could be impacted by physiological changes during the 13–24 h epoch after onset. In this study, a higher SI_d_ with greater variability was observed during the overall 24 h recording epoch and second 12 h epoch (13–24 h after onset), but not the first 12 h epoch, which suggested that SI_d_ may affect outcomes by mainly interfering vasogenic edema 13–24 h after onset. 

Our study has some limitations. Firstly, the population is relatively small due to the low incidence of LHI. Secondly, the ratios were collected less within the first 12 h epoch after onset due to the hospitalization delay of patients. Thirdly, our study lacked more hemodynamic parameters, such as PaCO2 and beat-to-beat data, which could have provided further insights into the underlying mechanisms. In order to validate our results, further studies are required across multiple centers, with a larger sample size. However, the limitations of this study do not obscure its strengths. Firstly, we focused on LHI, which is the very subtype of ischemic stroke with the worst outcome. We provided some specific evidence for intensive care in this population. Secondly, SI_s_ and SI_d_ are two simple indices derived from two readily available recordings during monitoring. It can be measured in emergencies or in the ICU without training from a neurologist, with usefulness in a clinical setting. Thirdly, we conducted serial measurements of HR and BP, revealing the temporal and variability pattern of the ratios. 

## 5. Conclusions

A higher and more fluctuated heart-rate-to-diastolic-blood-pressure ratio (SI_d_) independently correlated with increased risk of MBE and one-month death in LHI patients, especially during the second 12 h (13–24 h) epoch after onset. A higher heart rate-to-systolic blood pressure ratio (SI_s_) 13–24 h after onset was independently correlated with one-month death in LHI patients.

## Figures and Tables

**Figure 1 diagnostics-13-02506-f001:**
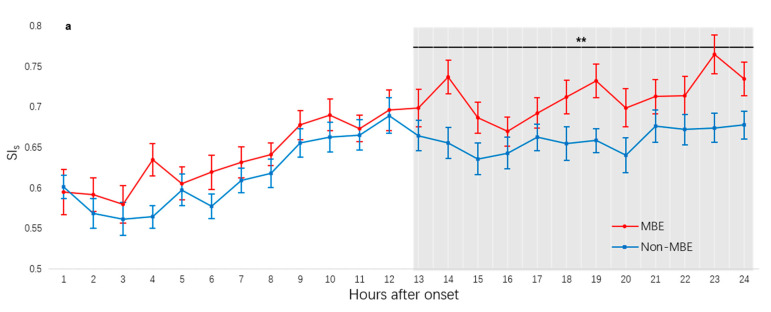
The 24 h temporal evolution of heart-rate-to-blood-pressure ratios in LHI patients with and without MBE. (**a**) Temporal evolution of SI_s_. (**b**) Temporal evolution of SI_d_. During 13–24 h after onset, in patients with MBE, SI_s_ and SI_d_ were higher than that in patients without MBE (*p* < 0.001 for SI_s_, *p* = 0.036 for SI_d_ by GEE method for MBE). LHI, large hemispheric infarction; MBE, malignant brain edema; SI, shock index; GEE, generalized estimating equation; heart-rate-to-blood-pressure ratios are assigned as SI_s_ and SI_d_ for systolic and diastolic pressures, respectively. * *p* < 0.05; ** *p* < 0.001.

**Figure 2 diagnostics-13-02506-f002:**
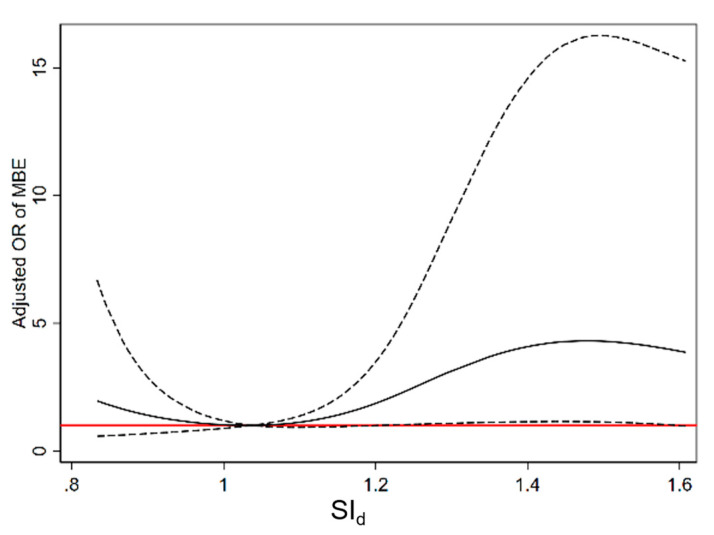
Restricted cubic spline regression analysis to explore the dose effect of mean level of SI_d_ 13–24 h after onset on MBE; solid black line stands for odds ratios, and the area between dotted black lines for 95% confidence intervals is based on standard errors. Red horizontal lines reflect grid lines when the odds ratio equals 1. Odds ratios were adjusted for age, body temperature, National Institute of Health stroke scale, ischemic area, hypertension, atrial fibrillation, dehydration treatment, and in-hospital infection. MBE, malignant brain edema; OR, odds ratio; SI, shock index; SI_d_ was calculated as heart rate divided by diastolic blood pressure.

**Table 1 diagnostics-13-02506-t001:** The Baseline Characteristics of Large Hemispheric Infarction Patients with and without Malignant Brain Edema.

	Total (*n* = 162)	With MBE (*n* = 46)	Without MBE (*n* = 116)	*p* Values
Demographics				
Age, years	70.31 ± 11.76	70.66 ± 11.11	70.13 ± 13.36	0.553
Male	71 (43.83)	18 (39.13)	53 (45.69)	0.448
Medical history				
Hypertension	79 (48.77)	21 (45.65)	58 (50)	0.618
Diabetes	26 (16.05)	10 (21.74)	16 (13.79)	0.214
Hyperlipidemia	6 (3.70)	2 (4.35)	4 (3.45)	0.785
Atrial fibrillation	65 (40.12)	18 (39.13)	47 (40.52)	0.871
Stenosis/occlusion of Intra/extracerebral Arteries	96 (59.26)	25 (54.35)	71 (61.21)	0.423
Clinical features				
GCS	11 (8–13)	11 (7–13)	11 (8.5–13.5)	0.480
NIHSS	16 (12–21)	16 (13–21)	16 (12–20)	0.218
Ischemic area ≥ 1/2 MCA territory	91 (56.17)	36 (78.26)	55 (47.41)	<0.001
Body temperature, °C	36.49 ± 0.37	36.48 ± 0.30	36.50 ± 0.50	0.779
Systolic blood pressure, mmHg	131.84 ± 1.19	131.80 ± 15.21	131.86 ± 15.23	0.981
Diastolic blood pressure, mmHg	75.60 ± 0.76	75.79 ± 1.51	75.53 ± 9.44	0.888
Heart rate, bpm	84.48 ± 1.20	87.29 ± 2.20	83.36 ± 1.42	0.139
TOAST classification				0.529
large-artery atherosclerosis	61 (37.65)	17 (36.96)	44 (37.93)	
cardioembolic	76 (46.91)	24 (52.17)	52 (44.83)	
small-artery occlusion	0 (0.0)	0 (0.0)	0 (0.0)	
acute stroke of other determined etiology	2 (1.23)	1 (2.17)	1 (0.86)	
stroke of underdetermined etiology	23 (14.20)	4 (8.70)	19 (16.38)	
In-hospital treatment				
Thrombolysis	26 (16.05)	8 (17.39)	18 (15.52)	0.770
Thrombectomy	47 (29.01)	18 (39.13)	29 (25.00)	0.074
Dehydration Therapy	144 (88.89)	44 (95.65)	100 (86.21)	0.085
In-hospital infection	125 (77.16)	38 (84.44)	87 (75.00)	0.197

All data were reported as mean ± standard deviation, median (interquartile range) or count (percentage) as appropriate; MBE, malignant brain edema; SD, standard deviation; IQR, interquartile range; GCS, Glasgow Coma Scale; NIHSS, National Institutes of Health Stroke Scale; MCA, Middle cerebral artery; TOAST, Trial of Org 10,172 in Acute Stroke Treatment.

**Table 2 diagnostics-13-02506-t002:** Calculated Parameters of Overall and Epoch-Based Heart-Rate-To-Blood-Pressure Ratios in Patients with and without Malignant Brain Edema.

Heart-Rate-to-Blood-Pressure Ratio Parameters, bpm/mmHg, Median (IQR)	Total (*n* = 162)	With MBE (*n* = 46)	Without MBE (*n* = 116)	*p* Values
Overall: 1–24 h				
SI_s_	0.64 (0.56–0.73)	0.68 (0.59–0.77)	0.62 (0.55–0.72)	0.110
SI_s_ max	0.85 (0.72–1.04)	0.89 (0.75–1.07)	0.84 (0.72–0.99)	0.163
SI_s_ min	0.47 (0.40–0.54)	0.48 (0.39–0.56)	0.46 (0.41–0.54)	0.551
SI_s_ range	0.38 (0.26–0.49)	0.42 (0.27–0.57)	0.37 (0.25–0.47)	0.192
SI_s_ SD	0.10 (0.07–0.15)	0.11 (0.09–0.16)	0.10 (0.07–0.14)	0.142
SI_s_ CV	0.17 (0.12–0.22)	0.18 (0.14–0.22)	0.16 (0.11–0.21)	0.138
SI_d_	1.14 (1.01–1.27)	1.20 (1.07–1.30)	1.13 (1.01–1.23)	0.079
SI_d_ max	1.52 (1.30–1.79)	1.56 (1.36–1.88)	1.50 (1.27–1.76)	0.085
SI_d_ min	0.82 (0.70–0.92)	0.83 (0.68–0.91)	0.81 (0.71–0.93)	0.971
SI_d_ range	0.74 (0.49–0.96)	0.82 (0.62–1.14)	0.71 (0.47–0.89)	0.029
SI_d_ SD	0.20 (0.14–0.28)	0.23 (0.17–0.30)	0.19 (0.13–0.26)	0.017
SI_d_ CV	0.18 (0.13–0.23)	0.21 (0.16–0.25)	0.17 (0.13–0.21)	0.018
Epoch #1: 1–12 h				
SI_s_	0.61 (0.54–0.71)	0.64 (0.57–0.76)	0.61 (0.54–0.68)	0.218
SI_s_ max	0.75 (0.65–0.88)	0.79 (0.69–0.95)	0.75 (0.62–0.87)	0.325
SI_s_ min	0.48 (0.42–0.57)	0.51 (0.43–0.59)	0.48 (0.42–0.56)	0.547
SI_s_ range	0.23 (0.13–0.38)	0.24 (0.13–0.44)	0.23 (0.13–0.36)	0.610
SI_s_ SD	0.09 (0.06–0.15)	0.10 (0.07–0.16)	0.09 (0.05–0.14)	0.184
SI_s_ CV	0.15 (0.10–0.22)	0.16 (0.10–0.25)	0.15 (0.09–0.21)	0.325
SI_d_	1.08 (0.94–1.22)	1.10 (0.96–1.27)	1.07 (0.93–1.21)	0.628
SI_d_ max	1.32 (1.13–1.56)	1.33 (1.13–1.59)	1.32 (1.13–1.51)	0.633
SI_d_ min	0.85 (0.72–0.95)	0.85 (0.69–0.95)	0.84 (0.75–0.95)	0.676
SI_d_ range	0.47 (0.27–0.69)	0.53 (0.27–0.72)	0.44 (0.27–0.65)	0.447
SI_d_ SD	0.17 (0.12–0.26)	0.19 (0.14–0.19)	0.16 (0.12–0.25)	0.155
SI_d_ CV	0.28 (0.20–0.40)	0.32 (0.20–0.45)	0.27 (0.19–0.37)	0.238
Epoch #2: 13–24 h				
SI_s_	0.66 (0.56–0.77)	0.70 (0.61–0.78)	0.64 (0.55–0.76)	0.161
SI_s_ max	0.78 (0.67–0.93)	0.80 (0.70–0.94)	0.77 (0.67–0.92)	0.181
SI_s_ min	0.56 (0.46–0.64)	0.54 (0.45–0.63)	0.59 (0.47–0.65)	0.200
SI_s_ range	0.24 (0.15–0.34)	0.24 (0.15–0.34)	0.23 (0.15–0.37)	0.659
SI_s_ SD	0.08 (0.06–0.13)	0.08 (0.06–0.15)	0.08 (0.06–0.12)	0.521
SI_s_ CV	0.13 (0.10–0.18)	0.14 (0.10–0.20)	0.13 (0.10–0.18)	0.682
SI_d_	1.17 (1.04–1.36)	1.26 (1.12–1.43)	1.13 (1.03–1.30)	0.028
SI_d_ max	1.42 (1.21–1.67)	1.54 (1.28–1.79)	1.40 (1.15–1.66)	0.057
SI_d_ min	0.95 (0.83–1.07)	1.01 (0.87–1.11)	0.93 (0.83–1.04)	0.139
SI_d_ range	0.48 (0.31–0.71)	0.56 (0.31–0.79)	0.45 (0.31–0.66)	0.108
SI_d_ SD	0.17 (0.13–0.25)	0.20 (0.16–0.30)	0.16 (0.12–0.25)	0.018
SI_d_ CV	0.15 (0.11–0.21)	0.17 (0.12–0.23)	0.14 (0.11–0.20)	0.089

MBE, malignant brain edema; IQR, interquartile range; SI, shock index; max, maximum; min, minimum; SD, standard deviation; CV, coefficient of variation; heart-rate-to-blood-pressure ratios are assigned as SI_s_ and SI_d_ for systolic and diastolic pressures, respectively; range was defined as the difference between maximum and minimum.

**Table 3 diagnostics-13-02506-t003:** Multivariate Analysis for Heart-Rate-To-Blood-Pressure Ratios Parameters in Large Hemispheric Infarction Patients.

Heart-Rate-to-Blood-Pressure Ratio Parameters	MBE				One-Month Death			
	OR (95%CI)	*p* Values	aOR *	*p* Values	OR (95%CI)	*p* Values	aOR ^§^	*p* Values
Overall: 1–24 h								
SI_s_	1.23 (0.95–1.59)	0.124	1.17 (0.88–1.55)	0.284	1.24 (0.95–1.62)	0.121	1.27 (0.94–1.70)	0.118
SI_s_ max	1.12 (0.97–1.30)	0.117	1.15 (0.98–1.35)	0.091	1.16 (1.00–1.35)	0.057	1.21 (1.02–1.44)	0.032
SI_s_ min	1.10 (0.79–1.54)	0.571	0.97 (0.67–1.41)	0.861	0.94 (0.66–1.35)	0.738	0.92 (0.62–1.35)	0.664
SI_s_ range	1.10 (0.95–1.27)	0.193	1.15 (0.98–1.35)	0.081	1.16 (1.00–1.35)	0.045	1.23 (1.03–1.46)	0.021
SI_s_ SD	1.69 (0.82–3.51)	0.158	1.76 (0.74–4.19)	0.205	1.89 (0.88–4.03)	0.100	2.06 (0.87–4.88)	0.102
SI_s_ CV	1.65 (0.71–3.87)	0.247	1.81 (0.66–5.00)	0.250	1.88 (0.78–4.55)	0.162	2.01 (0.74–5.47)	0.171
SI_d_	1.17 (0.99–1.39)	0.058	1.19 (0.98–1.44)	0.074	1.19 (1.00–1.42)	0.046	1.23 (1.01–1.50)	0.037
SI_d_ max	1.09 (1.00–1.18)	0.042	1.15 (1.04–1.28)	0.006	1.11 (1.02–1.21)	0.013	1.17 (1.05–1.29)	0.003
SI_d_ min	1.00 (0.81–1.23)	0.999	0.92 (0.73–1.16)	0.466	0.98 (0.79–1.21)	0.821	0.95 (0.75–1.21)	0.682
SI_d_ range	1.17 (1.06–1.30)	0.003	1.09 (1.00–1.18)	0.043	1.12 (1.03–1.22)	0.011	1.18 (1.06–1.31)	0.002
SI_d_ SD	2.21 (0.98–5.00)	0.056	3.72 (1.38–10.04)	0.010	2.29 (0.98–5.30)	0.054	3.45 (1.28–9.28)	0.014
SI_d_ CV	2.16 (0.82–5.71)	0.119	4.27 (1.33–13.69)	0.015	2.19 (0.80–5.96)	0.125	3.64 (1.14–11.69)	0.030
Epoch#1: 1–12 h								
SI_s_	1.14 (0.89–1.46)	0.310	1.04 (0.80–1.36)	0.750	1.08 (0.84–1.40)	0.536	1.02 (0.78–1.33)	0.872
SI_s_ max	1.08 (0.92–1.25)	0.351	1.03 (0.87–1.22)	0.755	1.09 (0.93–1.27)	0.283	1.07 (0.90–1.27)	0.442
SI_s_ min	1.09 (0.82–1.45)	0.542	1.01 (0.75–1.37)	0.946	0.90 (0.66–1.23)	0.512	0.85 (0.62–1.18)	0.332
SI_s_ range	1.06 (0.90–1.25)	0.515	1.03 (0.85–1.24)	0.759	1.14 (0.96–1.34)	0.126	1.15 (0.95–1.39)	0.145
SI_s_ SD	1.47 (0.84–2.56)	0.179	1.29 (0.68–2.48)	0.436	1.71 (0.96–3.03)	0.067	1.71 (0.90–3.26)	0.100
SI_s_ CV	1.43 (0.76–2.69)	0.267	1.32 (0.63–2.79)	0.462	1.83 (0.95–3.53)	0.073	1.88 (0.90–3.94)	0.093
SI_d_	1.03 (0.88–1.20)	0.720	0.99 (0.83–1.17)	0.873	1.05 (0.90–1.24)	0.520	1.02 (0.86–1.21)	0.788
SI_d_ max	1.03 (0.94–1.14)	0.542	1.01 (0.91–1.13)	0.845	1.06 (0.96–1.17)	0.217	1.06 (0.95–1.18)	0.299
SI_d_ min	0.98 (0.80–1.19)	0.806	0.93 (0.75–1.15)	0.503	0.95 (0.77–1.17)	0.618	0.92 (0.74–1.14)	0.436
SI_d_ range	1.04 (0.94–1.16)	0.430	1.04 (0.92–1.17)	0.551	1.09 (0.98–1.21)	0.112	1.10 (0.98–1.24)	0.115
SI_d_ SD	1.62 (0.84–3.10)	0.149	1.42 (0.68–2.97)	0.353	1.60 (0.84–3.07)	0.154	1.56 (0.77–3.18)	0.221
SI_d_ CV	1.78 (0.82–3.83)	0.142	1.62 (0.68–3.87)	0.277	1.70 (0.80–3.63)	0.170	1.70 (0.74–3.91)	0.209
Epoch#2: 13–24 h								
SI_s_	1.14 (0.93–1.41)	0.216	1.10 (0.86–1.41)	0.428	1.24 (1.00–1.54)	0.055	1.28 (1.00–1.65)	0.049
SI_s_ max	1.12 (0.97–1.30)	0.137	1.15 (0.97–1.38)	0.112	1.22 (1.04–1.42)	0.012	1.30 (1.08–1.56)	0.005
SI_s_ min	1.12 (0.86–1.46)	0.401	0.98 (0.73–1.33)	0.917	1.21 (0.92–1.60)	0.173	1.19 (0.88–1.61)	0.252
SI_s_ range	1.12 (0.94–1.34)	0.217	1.23 (1.00–1.52)	0.051	1.22 (1.02–1.47)	0.034	1.32 (1.07–1.63)	0.010
SI_s_ SD	1.35 (0.71–2.55)	0.362	1.50 (0.73–3.10)	0.269	1.39 (0.72–2.65)	0.325	1.50 (0.73–3.10)	0.267
SI_s_ CV	1.19 (0.58–2.41)	0.640	1.40 (0.64–3.08)	0.402	1.14 (0.55–2.35)	0.722	1.20 (0.56–2.61)	0.636
SI_d_	1.15 (1.00–1.31)	0.047	1.18 (1.00–1.39)	0.046	1.19 (1.03–1.37)	0.018	1.25 (1.06–1.48)	0.009
SI_d_ max	1.08 (1.00–1.17)	0.064	1.14 (1.03–1.26)	0.012	1.11 (1.02–1.20)	0.015	1.16 (1.05–1.28)	0.004
SI_d_ min	1.01 (0.99–1.04)	0.401	1.00 (0.97–1.03)	0.917	1.02 (0.99–1.05)	0.173	1.02 (0.99–1.05)	0.252
SI_d_ range	1.08 (0.98–1.18)	0.110	1.18 (1.05–1.32)	0.006	1.11 (1.01–1.22)	0.025	1.18 (1.05–1.32)	0.004
SI_d_ SD	2.30 (1.11–4.75)	0.025	3.16 (1.35–7.40)	0.008	1.97 (0.95–4.08)	0.068	2.38 (1.04–5.43)	0.039
SI_d_ CV	2.03 (0.88–4.71)	0.099	2.92 (1.13–7.51)	0.026	1.57 (0.68–3.65)	0.294	1.90 (0.76–4.76)	0.171

SI, shock index; SD, standard deviation; CV, coefficient of variation; OR, odds ratio; CI, confidence interval. Heart-rate-to-blood-pressure ratios are assigned as SI_s_ and SI_d_ for systolic and diastolic pressures, respectively. SD and CV values of SI_s_ and SI_d_ were entered into regression model as log-transformed value due to skewness distribution of data, while other SI_s_ and SI_d_ parameters were entered as ten-fold values (per 0.1-unit increase). * Adjusted by age, body temperature, NIHSS, ischemic area ≥ 1/2 MCA territory, hypertension, atrial fibrillation, dehydration treatment, and in-hospital infection; ^§^ Adjusted by age, body temperature, NIHSS, ischemic area ≥ 1/2 MCA territory, hypertension, atrial fibrillation, and in-hospital infection.

## Data Availability

The datasets used and/or analyzed in the current study are available from the corresponding author upon reasonable request.

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
