# Peer review of "Heart-Rate-to-Blood-Pressure Ratios Correlate with Malignant Brain Edema and One-Month Death in Large Hemispheric Infarction: A Cohort Study"

_diagnostics, 2023, doi:10.3390/diagnostics13152506_

Round 1
Reviewer 1 Report
Thank you for the opportunity to review this work. A few comments:
- It does not appear that your multivariable model adjusted for comorbidities. I did not see that the groups were otherwise matched. Why is this? Also, it appears that infarct size was different between the two groups. These all could account for the differences in the findings.
- Please describe in more detail the difference between the SI-dynamic and the SI-diastolic.
- Please indicate which variables were included in the multivariable model. If blood pressure alone, heart rate alone, and the the shock index were all included together, please indicate how collinearity was addressed.
- Is it possible that the SI changes DUE TO the edema and not the other way around as proposed by the study? Please include in the discussion.
Adequate
Author Response
Point 1: It does not appear that your multivariable model adjusted for comorbidities. I did not see that the groups were otherwise matched. Why is this? Also, it appears that infarct size was different between the two groups. These all could account for the differences in the findings.
Response 1:
Thank you very much for pointing this out. In this study, we adjusted two comorbidities which we think may influence outcomes the most-hypertension and atrial fibrillation (Method section-2.5 Statistical Analysis-Line 138-142; Table 3, Line 244-247). Due to limited event numbers (46 cases had MBE and 40 cases died within one-month), we regrated that we could not adjust more comorbidities according to ‘5-10 events per variable (EPV)’ rule since we also need to adjust some variables related to stroke features and treatments.
In terms of infarct size, we did find differences between groups. Therefore, we adjusted it in multivariable model both for MBE and one-month death (Method section-2.5 Statistical Analysis-Line 138-142; Table 3, Line 244-247).
Point 2: Please describe in more detail the difference between the SI-dynamic and the SI-diastolic.
Response 2:
Thank you very much for your advice. We have described more details about differences of SI dynamic in Line 172-185. Meanwhile, we have added more details in differences of SI-diastolic parameters among patients with and without MBE in Line 219-225, Line 226-234, and SI-diastolic parameters among patients died and survived within one month in Line 283-288, Line 289-295.
Point 3: Please indicate which variables were included in the multivariable model. If blood pressure alone, heart rate alone, and the the shock index were all included together, please indicate how collinearity was addressed.
Response 3:
Thank you very much for pointing this out. In multivariable models, we adjusted for age, body temperature, NIHSS, ischemic area (whether invisible hypodensity≥1/2 MCA territory of MCA), hypertension, atrial fibrillation, dehydration treatment, and in-hospital infection for MBE. We adjusted for age, body temperature, NIHSS, ischemic area (whether invisible hypodensity≥1/2 MCA territory of MCA), hypertension, atrial fibrillation, and in-hospital infection for one-month death. Considering collinearity, we didn’t enter blood pressure alone or heart rate alone into multivariable model. We described the above included variables in Method section-2.5 Statistical Analysis-Line 138-142, and Table 3-Line 244-247.
Point 4: Is it possible that the SI changes DUE TO the edema and not the other way around as proposed by the study? Please include in the discussion.
Response 4:
Thank you very much for pointing out this important issue. The median interval from onset to MBE development in our study was 36.3 hours, and 85% (39/46) of MBE was developed with an interval over 24 hours. For the 7 patients who developed MBE within 24h of onset, we didn’t apply the heart rate-to-blood pressure ratios collected after MBE development for analysis, in order to ensure the appropriate sequential order of studied parameters and outcomes. For the rest patients we included all collected heart rate-to-blood pressure ratios for analysis. Therefore, it is much more likely that the MBE development was influenced by ratio changes and not the other way around, since for each individual patient, the studied ratios were collected strictly before MBE occurred.
We have included this in discussion section (Line 315-325). Additionally, we added several descriptions about excluding heart rate-to-blood pressure ratios collected after MBE development for analysis in Method
Reviewer 2 Report
Dear Authors,
discriminating the outcome of patients with large stroke is important for both sides, physicians and patients/families. heart rate to blood pressure is a simple and reliable method, which is suitable at any intensive care station. the manuscript adds the evaluation of this ratio In such patients and helps to early identify patients with high risk for unfortunate outcome.
Abstract: too long, please shorten. line 18, onset: please mention once ‘onset of symptoms’ – if you mean that. line 23: please also give n (in addition to %). lines 25-32 are heard to read and understand, please just one correlation in one sentence.
Introduction: please explain cytotoxic and vasogenic edema here and what is known about them.
Methods: here and in the whole manuscript you use a lot of abbreviations, this makes reading not easier. line 104, mRS, please explain also this one, probably modified rankin scale.
Results: Again, lots of abbreviations. And, throughout the whole ms spaces are missing after words. please check that to improve readability. line 142: please mention once ‘onset of symptoms’ if you mean that. line 147, why not 162 patients? Table 1: n, %, IQR etc. should be explained below, not in each line. Figure 1: axis labels are hard to read, why don’t you put the right graph below? description ‘edema’ probably means ‘MBE’; summarized date should be expressed as mean+-sd, not just as one point. Table 3: cv and sd not explained. data on heart rate variability (HRV) would be interesting here: HRV in hour 1-12 and 13-24: how does it correlate with the endpoints?
Discussion: lines 303ff and 322ff, baroflex and sepsis, please shorten these paragraphs as they don’t directly relate to the data. lines 351ff: limitations at first.
English is ok, minor changes will improve readability.
Author Response
Point 1:
Abstract: too long, please shorten.
Response 1:
Thank you very much for pointing this out. We have shortened the abstract.
Point 2:
line 18, onset: please mention once ‘onset of symptoms’ – if you mean that.
Response 2:
Apologies for our negligence. We have added ‘onset of symptoms’ in Abstract (Currently Line 18-19)
Point 3:
line 23: please also give n (in addition to %).
Response 3:
Apologies for our negligence. We have given n in addition to % in Line 23. To note, the denominator for one-month death was 158 because 4 patients didn’t respond to follow-up interview.
Point 4:
lines 25-32 are heard to read and understand, please just one correlation in one sentence.
Response 4:
Apologies for causing confusion. We have modified our expression in order to clarify the correlations more clearly.
Point 5:
Introduction: please explain cytotoxic and vasogenic edema here and what is known about them.
Response 5:
Thank you for pointing this out. We have explained and described cytotoxic and vasogenic edema in Line 67-73.
Point 6:
Methods: here and in the whole manuscript you use a lot of abbreviations, this makes reading not easier. line 104, mRS, please explain also this one, probably modified rankin scale.
Response 6:
Thank you for pointing this out. We have reduced the using of abbreviation in the whole manuscript, and only used abbreviation for studied parameters (SI), outcome (MBE) and those which are well established (for example NIHSS).
We added ‘modified Rankin scale’ for full name of mRS (Currently in Line 122).
Point 7:
Results: Again, lots of abbreviations. And, throughout the whole ms spaces are missing after words. please check that to improve readability. line 142: please mention once ‘onset of symptoms’ if you mean that.
Response 7:
Apologies for our negligence. We have reduced the using of abbreviation and added missing spaces after words. We added ‘onset of symptoms’ to make the sentence as ‘The median delay from onset of symptoms to admission was 5 hours’ (Currently Line 160).
Point 8: line 147, why not 162 patients?
Response 8:
In our study, a total of 4 patients at one month and 7 patients at three months were lost to follow-up. So only 158 patients fulfilled one-month follow up, and 155 patients fulfilled three-month follow up (Currently in Line 164-165).
Point 9:
Table 1: n, %, IQR etc. should be explained below, not in each line.
Response 9:
Thank you for pointing this out. We explained n, %, IQR etc. below the table (Table 1, Line 205-206).
Point 10:
Figure 1: axis labels are hard to read, why don’t you put the right graph below? description ‘edema’ probably means ‘MBE’; summarized date should be expressed as mean+-sd, not just as one point.
Response 10:
Apologies for our negligence. We have moved the right graph below, changed description ‘edema’ into MBE, and summarized data as mean+-sd (Figure 1, Line 186).
Point 11:
Table 3: cv and sd not explained.
Response 11:
Apologies for our negligence. We have added explanation of SD and CV (Table 3, Line 239).
Point 12:
data on heart rate variability (HRV) would be interesting here: HRV in hour 1-12 and 13-24: how does it correlate with the endpoints?
Response 12:
Thank you very much for your constructive advice.
We calculated HRV in hour 1-12 and 13-24 after onset of symptom in our study, and conducted multivariable logistic regression for MBE and one-month death by adjusting the same covariables used in the manuscript. No significant correlation was found between HRV within 1-12h post-stroke or HRV within 13-24h post-stroke with MBE. Greater HRV during 13-24 hour after onset was significantly correlated with one-month death after adjustment (OR 1.08, 95%CI 1.01-1.15, p=0.021). However, this correlation was opposite to previous report (for example, DOI: 10.1111/ijs.12573) that reduced HRV is associated with stroke severity, early and late complications, dependency, and mortality. Anyhow, this is a very interesting point of view, which could be investigated in future larger sample studies, and needs another manuscript to discuss with. Again, thank you very much for this brilliant idea.
Point 13:
Discussion: lines 303ff and 322ff, baroflex and sepsis, please shorten these paragraphs as they don’t directly relate to the data. lines 351ff: limitations at first.
Response 13:
Thank you very much for your advice. We have shortened these paragraphs (Line 366-387, Line 388-405), deleting some content which didn’t directly relate to our result. Meanwhile, we put limitations before strengths (Cur
Round 2
Reviewer 1 Report
Thank you for the opportunity to review this revision. My concerns appear to have been mostly addressed with the exception of one comment:
- Please clarify where you are referring to SI diastolic and where you are referring to SI dynamic. I would suggest differentiating them using something like SIdia and SIdy or something similar. It is very difficult to follow.
slightly unusual word choice (e.g. "epoch" instead of period or time). Otherwise seems fine.
Author Response
Point 1: Please clarify where you are referring to SI diastolic and where you are referring to SI dynamic. I would suggest differentiating them using something like SIdia and SIdy or something similar. It is very difficult to follow.
Response 1:
Apologize for our negligence. In this manuscript, all SId referred to SI diastolic, as was defined in Method section (Line 103-107). We described the dynamic change, or temporal evolution in another word, of SIs and SId for 24 hours after onset of symptoms in ‘Result section-3.2 Temporal Evolution of Heart Rate-to-Blood Pressure Ratios’ (Line 167-185). We didn’t apply any abbreviation for a specific ‘SI dynamic’. We are sorry for causing confusion. Therefore, we changed the expression where the word ‘dynamic’ was used (Line 345, Line 429). In Line 345, we apologize that there was a misusing of the word ‘dynamic’, so we deleted it here. In Line 429, we changed the word ‘dynamic’ into ‘temporal’ to reduce confusion.
Point 2: slightly unusual word choice (e.g. "epoch" instead of period or time). Otherwise seems fine.
Response 2:
Thank you for pointing this out. We have changed the word choice (Line 110, Line 175, Line 199, Line 219, Line 224, Line 227, Line 279, Line 283, Line 295, Line 329, Line 332, Line 333, Line 408, Line 413, Line 414), by changing ‘period’ into ‘epoch’ as advised.
